# Oral Administration of Lipopolysaccharide Enhances Insulin Signaling-Related Factors in the KK/Ay Mouse Model of Type 2 Diabetes Mellitus

**DOI:** 10.3390/ijms24054619

**Published:** 2023-02-27

**Authors:** Kazushi Yamamoto, Masashi Yamashita, Masataka Oda, Vindy Tjendana Tjhin, Hiroyuki Inagawa, Gen-Ichiro Soma

**Affiliations:** 1Control of Innate Immunity, Technology Research Association, Takamatsu 761-0301, Japan; 2Research Institute for Healthy Living, Niigata University of Pharmacy and Applied Life Sciences, Niigata 956-0841, Japan

**Keywords:** glucose tolerance, adipose tissue, insulin resistance, lipopolysaccharide, oral administration

## Abstract

Lipopolysaccharide (LPS), an endotoxin, induces systemic inflammation by injection and is thought to be a causative agent of chronic inflammatory diseases, including type 2 diabetes mellitus (T2DM). However, our previous studies found that oral LPS administration does not exacerbate T2DM conditions in KK/Ay mice, which is the opposite of the response from LPS injection. Therefore, this study aims to confirm that oral LPS administration does not aggravate T2DM and to investigate the possible mechanisms. In this study, KK/Ay mice with T2DM were orally administered LPS (1 mg/kg BW/day) for 8 weeks, and blood glucose parameters before and after oral administration were compared. Abnormal glucose tolerance, insulin resistance progression, and progression of T2DM symptoms were suppressed by oral LPS administration. Furthermore, the expressions of factors involved in insulin signaling, such as insulin receptor, insulin receptor substrate 1, thymoma viral proto-oncogene, and glucose transporter type 4, were upregulated in the adipose tissues of KK/Ay mice, where this effect was observed. For the first time, oral LPS administration induces the expression of adiponectin in adipose tissues, which is involved in the increased expression of these molecules. Briefly, oral LPS administration may prevent T2DM by inducing an increase in the expressions of insulin signaling-related factors based on adiponectin production in adipose tissues.

## 1. Introduction

Lipopolysaccharide (LPS) is a glycolipid found in the outer membrane of Gram-negative bacteria. When LPS is injected into the body, it binds to Toll-like receptors (TLR4) in vivo, even at very low doses. This binding leads to the production of pro-inflammatory cytokines and, in turn, causes a strong systemic inflammatory response. This condition is called a cytokine storm state, which can induce shock symptoms such as fever and diarrhea and may lead to death [1,2,3,4]. Since the discovery of LPS, it has been called an endotoxin [5] and has been used as an inflammation-inducing substance for stimulating immune cells in vitro and inflammation models in animals (in vivo) by injection. Moreover, bacterial translocation has been observed in persistent inflammatory lesions in the intestinal tract and periodontal tissues, and persistent invasion of bacteria and LPS in living organisms induces systemic inflammation [6]. Based on these findings, LPS is considered a cause of chronic inflammatory diseases, including lifestyle-related diseases [7]. Specifically, many studies have reported that LPS is a causative agent of type 2 diabetes mellitus, one of the major lifestyle diseases.

Type 2 diabetes mellitus is a chronic inflammatory disease of adipose tissues and is characterized by impaired glucose tolerance and insulin resistance. Several studies have suggested that LPS induces the onset of type 2 diabetes mellitus. For example, LPS injection was reported to decrease the protein and mRNA expression of Glucose transporter type 4 (Glut 4), the primary transporter for glucose uptake in adipose tissues, and induces symptoms of type 2 diabetes mellitus such as increased fasting blood glucose, impaired glucose tolerance, and insulin resistance [1,2,3,4]. However, in these reports, LPS was injected intraperitoneally or intravenously to induce pathological models of systemic inflammation deliberately. LPS exists within healthy individuals, yet inflammation was not induced [8].

A hundred trillion bacteria reside in the mucosa of animals, and approximately half of those are Gram-negative bacteria that contain LPS on their cell wall. Thus, LPS is permanently present in the mucosa and should not be considered toxic to living beings. Indeed, our group found that oral LPS administration does not induce inflammation, unlike its injections [9,10]. Furthermore, oral LPS administration was found to suppress inflammation induced by a high-fat diet, prevent dementia, and suppress atherosclerosis in senescence-accelerated mice (SAM-P8) and ApoE-deficient atherosclerotic mice fed a high-fat diet [11,12]. These findings suggest that the physiological role of LPS differs from that of inflammation exacerbation by injection. Therefore, we propose that orally administered LPS should not be viewed as an endotoxin that induces inflammation but as a potentially beneficial substance that merits further studies.

Hence, this study aimed to establish this proposition by investigating the effect of oral LPS administration on type 2 diabetes mellitus using the KK-Ay mouse model, which is the standard animal model of type 2 diabetes mellitus [13,14]. As a result, we found that oral LPS administration improved fasting blood glucose levels and glucose tolerance index (HOMA-IR) without exacerbating inflammatory markers in KK/Ay mice with diabetic symptoms. In addition, for the first time, we found that adiponectin, an adipokine important for regulating glucose metabolic function, is induced in adipose tissues.

## 2. Results

### 2.1. Verification of the Development of Type 2 Diabetes Mellitus in KK/Ay Mice

To investigate the effect of oral LPS administration on type 2 diabetes mellitus, KK/Ay mice, a standard model for this analysis, were used because they exhibit impaired glucose tolerance and insulin resistance. Initially, an oral glucose tolerance test (OGTT) and measurement of blood glucose parameters were performed to confirm the condition of KK/Ay mice and compared it with the condition of non-type 2 diabetes mellitus model C57BL/6 mice.

The OGTT results show that blood glucose levels were significantly increased in KK/Ay mice compared with those in C57BL/6 mice at 0, 30, 60, 120, and 180 min after oral glucose administration (Figure 1a). The area under the curve (AUC) of the OGTT was also significantly increased in KK/Ay mice compared with that in C57BL/6 mice (Figure 1b). In addition, the blood glucose parameters, such as fasting blood glucose level and hemoglobin A1C (HbA1c) test results, were significantly increased in KK/Ay mice compared with those in C57BL/6 mice (Figure 1c,d). Furthermore, blood insulin and HOMA-IR, an indicator of insulin resistance, were significantly increased in KK/Ay mice compared with those in C57BL/6 mice (Figure 1e,f).The abnormal glucose tolerance and insulin resistance results from these tests confirmed that KK/Ay mice had type 2 diabetes mellitus at the start of the study.

### 2.2. Oral LPS Administration Suppresses Type 2 Diabetes Mellitus in KK/Ay Mice

KK/Ay mice with type 2 diabetes mellitus were divided into two groups: the LPS (−) group received distilled water, and the LPS (+) group received distilled water containing LPS. After 8 weeks, the effects of oral LPS administration on insulin resistance and glucose intolerance in KK/Ay mice were examined by performing an OGTT and measuring blood glucose parameters. In the OGTT, blood glucose levels at 60, 120, and 180 min after oral glucose administration were significantly lower in the LPS (+) group than in the LPS (−) group (Figure 2a), and the AUC of the OGTT was also significantly lower in the LPS (+) group than in the LPS (−) group (Figure 2b). Regarding blood glucose parameters, the fasting blood glucose showed a decreasing trend in the LPS (+) group compared with that in the LPS (−) group, and HbA1c decreased significantly in the LPS (+) group when compared with that in the LPS (−) group (Figure 2c,d). In addition, blood insulin levels appear to decrease in the LPS (+) group compared with those in the LPS (−) group, and HOMA-IR, a marker of insulin resistance, decreased significantly in the LPS (+) group compared with that in the LPS (−) group (Figure 2e,f). These results suggest that oral LPS administration has an ameliorating or inhibitory effect on insulin resistance and glucose intolerance in type 2 diabetic KK/Ay mice.

Next, the OGTT and blood glucose parameters at the start (week 0) and end (week 8) of the study were compared to determine whether oral LPS administration had an ameliorating or inhibitory effect on type 2 diabetes mellitus. The AUC of the OGTT increased significantly in the LPS (−) group at the end of the study compared with that at the start of the study; however, no significant difference was found before and after the start of the study in the LPS (+) group (Figure 3a). Fasting blood glucose levels were not statistically different between the LPS (−) and LPS (+) groups; however, the fasting blood glucose level increased approximately 1.2-fold in the LPS (−) group, whereas it was almost unchanged in the LPS (+) group (Figure 3b). HbA1c increased significantly in both the LPS (−) and LPS (+) groups at the end of the study compared with that at the beginning of the study. However, the increase in HbA1c was approximately 1.6-fold in the LPS (−) group, whereas it was approximately 1.3-fold in the LPS (+) group, with the rate of increase also being lower in the LPS (+) group (Figure 3c). Blood insulin levels in both the LPS (−) and LPS (+) groups also increased significantly at the end of the study compared with that at the beginning of the study. The LPS (−) group had a 2.5-fold increase in blood insulin level, whereas the LPS (+) group had a 1.5-fold increase, with the rate of increase being lower in the LPS (+) group (Figure 3d). HOMA-IR increased significantly in the LPS (−) group at the end of the study compared with that at the beginning of the study, and no significant difference was observed in the LPS (+) group before and after the study (Figure 3e). These results indicate that oral LPS administration suppressed the progression of insulin resistance and glucose intolerance in type 2 diabetic KK/Ay mice, suggesting that oral LPS administration has a suppressive effect on type 2 diabetes mellitus.

### 2.3. Effects of Oral LPS Administration on Adipose Tissues

The analysis of body weight and adipose tissue changes in mice with and without LPS administration showed no significant changes in body weight, mesenteric adipose tissue weight, perirenal adipose tissue weight, peritesticular adipose tissue weight, or total adipose tissue weight (Figure 4a–e). On the contrary, the size of cells in adipose tissues was significantly reduced by oral LPS administration (Figure 4f,g).

### 2.4. Effects of Oral LPS Administration on the Expression Levels of Insulin Signaling-Related Factors in Adipose Tissues of KK/Ay Mice

The adipose tissue is one of the tissues that plays a crucial role in type 2 diabetes mellitus, specifically its insulin signaling. The elevated insulin signaling in adipose tissues is involved in suppressing the progression of type 2 diabetes mellitus. Therefore, we hypothesized that the expression levels of insulin signaling-related factors are upregulated in adipose tissues of KK/Ay mice, in which blood insulin resistance suppression and blood glucose intolerance by oral LPS administration were observed. The expression levels of insulin signaling-related factors in adipose tissues indicated that mRNA and protein expression levels of Glut4, which is involved in insulin-mediated glucose uptake, were much elevated in the LPS (+) group than in the LPS (−) group (Figure 5a,b). In addition, the expression level of insulin receptor (*Ir*), which is involved in insulin signaling to upregulate Glut4 expression, showed an increasing trend in the LPS (+) group compared with that in the LPS (−) group (Figure 5c). Moreover, the expression levels of thymoma viral proto-oncogene (*Akt*) and insulin receptor substrate 1 (*Irs1*) were significantly elevated in the LPS (+) group compared with those in the LPS (−) group (Figure 5d,e). These results suggest that the expression levels of insulin signaling-related factors in adipose tissues are upregulated in KK/Ay mice, in which glucose intolerance and insulin resistance were suppressed by oral LPS administration.

### 2.5. Adiponectin Expression in Adipose Tissues of KK/Ay Mice by Oral LPS Administration

The induction of the expression levels of insulin signaling-related factors in adipose tissues is thought to involve adiponectin, a cytokine synthesized uniquely by adipocytes. Therefore, we hypothesized that when the expression levels of insulin signaling-related factors in adipose tissues were increased by oral LPS administration, adiponectin expression in adipose tissues would be also induced.

The adiponectin gene expression and protein levels in adipose tissues were measured. The results show that adiponectin mRNA expression was significantly elevated in the LPS (+) group compared with that in the LPS (−) group (Figure 6a). In addition, the protein level of adiponectin was also significantly elevated in the LPS (+) group compared with that in the LPS (−) group (Figure 6b). Based on these results, we hypothesized that oral LPS administration induces adiponectin expression and upregulates the expression levels of insulin signaling-related factors in adipose tissues. Further investigations on adiponectin showed that the mRNA expression levels of adiponectin receptors (*Adipor1* and *Adipor2*), which are located on the reaction pathway of adiponectin, were significantly increased in the LPS (+) group compared with that in the LPS (−) group (Figure 6c,d). Adiponectin directly induces increased Glut4 expression; however, whether it directly induces the expression levels of *Ir*, *Irs1*, and *Akt2* is unclear. Therefore, to investigate whether adiponectin directly induces an increase in the expression levels of insulin signaling-related factors, we stimulated 3T3-L1 adipocytes in vitro with adiponectin (Figure 7a). The results showed that the mRNA expression of *Glut4* tended to increase, and the expression levels of *Ir*, *Irs1*, and *Akt2* mRNAs in 3T3-L1 adipocytes significantly increased upon adiponectin stimulation (Figure 7b–e). Thus, the upregulation of *Glut4*, *Ir*, *Irs1*, and *Akt2* observed in the adipose tissues of KK/Ay mice orally treated with LPS may be due to adiponectin induction by LPS.

Based on these findings, we speculated that oral LPS administration induces the expression of adiponectin in adipose tissues and that adiponectin directly induces the upregulation of insulin signaling-related factors in adipose tissues by oral LPS administration.

## 3. Discussion

Enteral LPS is responsible for the onset of obesity-related type 2 diabetes mellitus [15,16,17,18]. This perception is based on the fact that overeating and high-fat diets increase the amount of enteral LPS transferred into the blood [18,19]. In this study, glucose intolerance and insulin resistance are induced in a mouse model receiving continuous infusion of LPS using a subcutaneously implanted infusion pump [16]. However, previous reports have also demonstrated that intestinal LPS is incorporated into chylomicrons and transferred into blood and that chylomicron LPS is barely involved in disease induction [20,21,22]. Based on the reports, evidence is insufficient to support the hypothesis that LPS is the cause of type 2 diabetes mellitus. In this study, oral LPS administration suppressed the onset of type 2 diabetes mellitus in KK/Ay mice by increasing the expression levels of insulin signaling-related factors in adipose tissues.

In our previous study, oral LPS administration to ApoE KO mice fed a high-fat diet and P8 (SAMP8), a mouse model of accelerated aging, reduced insulin resistance and AUC during a glucose tolerance test. This effect was observed at higher oral LPS doses (1 mg > 0.3 mg/kg/day) [11,12]. Based on these results, the LPS dose to be given orally in this study was set at 1 mg/kg/day.

KK/Ay mice were used as experimental animals in this study. These mice were developed by crossing KK mice with C57BL/6-Ay mice carrying the *Agouti gene* (*Ay*) [23,24]. They exhibit hyperglycemia, high HbA1c levels, insulin resistance, and impaired glucose tolerance, so they are widely used as a standard mouse model of type 2 diabetes mellitus to investigate substances that are beneficial for type 2 diabetes mellitus and analyze the mechanism of action of type 2 diabetes mellitus [13,14]. Therefore, we determined that the KK/Ay mouse model was appropriate for this study to clarify the effects of oral LPS administration on type 2 diabetes mellitus.

The initial state of KK/Ay mice was compared with that of naïve C57BL/6 mice, which were used as the non-type 2 diabetes mellitus model. At the beginning of the study, measurements of OGTT and blood parameters showed that KK/Ay mice had more advanced glucose tolerance and insulin resistance than C57BL/6 mice (Figure 1). Most of the studies using KK/Ay mice have concluded that KK/Ay mice had type 2 diabetes mellitus based on these results [25,26,27]. Therefore, we can say that KK/Ay mice had type 2 diabetes mellitus at the beginning of this study. The results of OGTT and measurements of blood parameters performed after 8 weeks of oral LPS administration were consistent with previous results from Apo E KO mice and SAMP8 [11,12]. Oral LPS administration led to a significant increase in the AUC of OGTT, fasting blood glucose, HbA1c level, and insulin resistance (Figure 2). These results suggest that oral LPS administration has an ameliorating or inhibitory effect on KK/Ay mice with type 2 diabetes mellitus. In this study, the effect of oral LPS administration on type 2 diabetes mellitus was further verified by comparing OGTT and blood parameters before and after oral LPS administration (Figure 3). The results were novel in that oral LPS administration has an inhibitory effect on type 2 diabetes mellitus. On the contrary, in a previous study, we reported that oral LPS administration to pre-diabetic (type 2) humans reduced fasting blood glucose and HbA1c levels compared with those before oral LPS administration, which are slightly different from the results of the present study [28]. However, in this previous study, LPS was orally administered in combination with Salacia tea, which has hypoglycemic effects. The oral LPS administration method was different from the method employed in the present study, in which only LPS was administered orally. This report also indicates that oral LPS administration enhanced the hypoglycemic effect of Salacia tea. Thus, LPS, when administered orally alone, has an inhibitory effect on type 2 diabetes mellitus; however, when taken in combination with other active ingredients or foods, it appeared to have an ameliorating effect on type 2 diabetes mellitus.

Many studies have shown that the adipose tissue is one of the key tissues involved in the suppression of type 2 diabetes mellitus [29,30,31,32,33,34,35,36]. The mRNA expression levels of insulin signaling-related factors such as *Ir*, *Irs1*, *Akt*, and *Glut4* declined in adipose tissues in the case of type 2 diabetes mellitus [31,34,36]. Furthermore, mice with adipose tissue-specific knockout of *Glut4* mRNA have abnormal glucose tolerance and insulin resistance, even though the expression levels of insulin signaling-related factors in other tissues were comparable to those of wild-type healthy mice [37]. A study reported that glucose intolerance was also observed in mice with adipose tissue-specific knockout of *Ir* [38]. Conversely, glucose intolerance and insulin resistance were suppressed in mice with adipose tissue-specific enhanced expression of Glut4 [39,40]. These studies have suggested that increased expression levels of insulin signaling-related factors in adipose tissues are crucial in the suppression of type 2 diabetes mellitus. This study revealed that gene and protein expression levels of insulin signaling-related factors were elevated in KK/Ay mice, whose glucose intolerance and insulin resistance were suppressed by oral LPS administration (Figure 5). Therefore, these results suggest that oral LPS administration increases the expression levels of insulin signaling-related factors in adipose tissues of KK/Ay mice and suppresses insulin resistance and glucose tolerance.

The expression levels of insulin signaling-related factors in adipose tissues are thought to be regulated by cytokines produced by adipose tissues [32,41]. Among these factors, adiponectin is a cytokine produced specifically by adipocytes and induces the expression levels of insulin signaling-related factors in adipose tissues [42,43]. For example, in vitro, adiponectin directly induces an increase in Glut4 mRNA and protein expression in adipocytes [43]. In vivo, mice with higher expression levels of adiponectin have increased *Glut4* expression levels in adipose tissues and suppressed glucose intolerance and insulin resistance [42]. In this study, the expression of adiponectin in adipose tissues of KK/Ay mice, which showed suppression of type 2 diabetes mellitus by oral LPS administration, was increased, and the adiponectin receptor, a molecule in the pathway for adiponectin to induce increased the gene expression levels of insulin signaling-related factors, was also upregulated (Figure 6). Furthermore, in vitro adiponectin stimulation studies on 3T3–L1 adipocytes confirmed that adiponectin directly induces an increase in *Glut4* expression, consistent with previous reports (Figure 7). In addition, adiponectin was demonstrated to directly induce increased mRNA expression levels of *Ir*, *Irs1*, and *Akt2* (Figure 7). These results suggest that adiponectin induced by oral LPS administration upregulates the expression levels of insulin signaling-related factors in adipose tissues (Figure 8).

LPS injection not only induces type 2 diabetes mellitus by decreasing insulin signaling in adipose tissues but also induces increased expression levels of inflammatory cytokines such as interleukin 1 beta (IL-1β), IL-6, monocyte chemotactic protein 1 (Mcp1), and tumor necrosis factor alpha (TNFα) and induces weight changes, hepatotoxicity, and dyslipidemia. However, oral LPS administration did not induce the mRNA expression levels of *IL-1b*, *IL-6*, *IL-12b*, *Mcp1*, or *TNFα* in adipose tissues of KK/Ay mice (Appendix A). In addition, it increased the expression levels of insulin signaling-related factors in adipose tissues and suppressed type 2 diabetes mellitus.

No effects were observed on body weight, adipose tissue weight, and blood Alanine transaminase (ALT) and aspartate transaminase (AST) levels, markers of hepatotoxicity (Figure 4 and Appendix A). Furthermore, the results of dyslipidemia markers, such as blood triglyceride (TG), total cholesterol (TC), low-density lipoprotein (LDL), and high-density lipoprotein (HDL), were in agreement with our previous reports [11,12], showing a significant decrease in TC and LDL, consequently a suppressive effect on dyslipidemia (Appendix A). Dyslipidemia is suppressed by adiponectin [44], and the results of this study revealed that this effect may be related to the increase in adiponectin levels in adipose tissues induced by oral LPS administration. Furthermore, the results corresponded with our initial proposition that oral LPS administration induces an entirely different effect on organisms compared with LPS injection; thus, we proposed that the conventional idea that LPS is involved in the development of type 2 diabetes mellitus should be revised.

The pathway by which orally administered LPS affects adipose tissues is unknown. Lu et al. found that mice with small intestine-specific *TLR4* knockout had impaired glucose tolerance [45], suggesting that orally administered LPS acts starting from *TLR4* in the small intestine in vivo. Furthermore, repeated low-dose LPS stimulations mimicking oral LPS administration in vitro do not induce adiponectin expression in 3T3-L1 adipocytes [46]. Thus, we hypothesized that orally administered LPS induces adiponectin expression through an indirect effect on adipose tissues. As possible mediators involved in this signaling process, we found that the membrane-bound Csf1 of blood monocytes is one of the second signaling molecules that transmits the signal of orally administered LPS to distal tissues [10]. In the future, we would like to clarify the mechanism of the inhibitory effect of orally administered LPS on type 2 diabetes mellitus, including analysis of the second signal in the control of adipose tissues.

## 4. Materials and Methods

### 4.1. Animal Experiments

Male KK/Ay mice, aged 7 weeks, were purchased from CLEA Japan (Tokyo, Japan), and male C57BL/6 mice, aged 7 weeks, were purchased from SLC Japan (Hamamatsu, Japan) and maintained in a temperature- and humidity-controlled room under a 12 h light/dark cycle with unrestricted access to food and water. A mouse diet (low-fat diet (LFD); 16.1 kJ/g, 4.3% *w/w* fat and 0.005% *w*/*w* cholesterol; D12450B) was purchased from Research Diets, Inc. (New Brunswick, NJ, USA). All mice were acclimated for 1 week while fed on an acclimation diet (CE-2; CLEA Japan, Tokyo, Japan) and drank sterilized distilled water. KK/Ay mice were assigned to the LPS (+) and LPS (−) groups and fed an LFD for 8 weeks. C57BL/6 mice were assigned to one group (control group) and fed an LFD for 8 weeks. Purified LPS derived from *P*. *agglomerans* (obtained from Macrophi Inc., Kagawa, Japan) was dissolved in sterilized distilled water and applied at 1 mg/kg body weight (BW)/day. The LPS dose was estimated from previous in vivo studies, in which the dose required to achieve preventive effects was determined (0.1 ± 1 mg/kg BW/day) [9,10,11,12]. The drinking water was replaced weekly, and the concentration of LPS was adjusted according to the average BW and amount of water consumption. We previously confirmed that LPS degradation in drinking water in a week was not significant [11,12]. At the end of the experiment, the KK/Ay mice were anesthetized under isoflurane vapor and euthanized by decapitation. Whole blood was collected, and a portion was stored at −80 °C until assays were performed. The rest was centrifuged (2000× *g* for 20 min at 4 °C), and the resulting plasma or serum (supernatant) was stored at −80 °C until assays were performed. Mesenteric, perinephric, and epididymal adipose tissues were collected, weighed, and stored at −80 °C until assays were performed. At the end of the experiment, the C57BL/6 mice were anesthetized under isoflurane vapor and euthanized by decapitation. Serum or plasma was collected in the same manner as with KK/Ay mice and stored at −80 °C until assays were performed.

The animal experiments were reviewed and approved by the Animal Care and Use Committee of the Control of Innate Immunity (approval number: CIITRA 02–08, CIITRA 02–09). This experiment was conducted according to the Law for the Humane Treatment and Management of Animal Standards Relating to the Care and Management of Laboratory Animals and Relief of Pain (Ministry of the Environment, Tokyo, Japan), the Fundamental Guidelines for Proper Conduct of Animal Experiments and Related Activities in Academic Research Institutions (Ministry of Education, Culture, Sports, Science and Technology, Tokyo, Japan), and the Guidelines for Proper Conduct of Animal Experiments (Science Council of Japan).

### 4.2. OGTTs

Mice were fasted overnight and subjected to an OGTT by oral glucose administration (gavage with 2 g of D-glucose/kg BW). Blood samples were collected from the tail vein, and blood glucose levels were monitored using an Accu-Chek Aviva blood glucose meter with Accu-Chek Aviva test strips (Roche Diagnostics K.K., Tokyo, Japan) at 0, 30, 60, 120, and 180 min after glucose loading. The AUC was calculated using the trapezoid rule.

### 4.3. Biochemical Analyses of Serum or Plasma, Whole Blood, and Epididymal ADIPOSE Tissues

TG, TC, LDL, HDL, glucose, AST, and ALT levels in the serum or plasma were measured using commercial enzyme kits (Wako Pure Chemical, Osaka, Japan) according to the manufacturer’s protocol. Insulin was determined using ELISA kits (Shibayagi, Shibukawa, Japan). HbA1c was measured by Oriental Yeast CO., Ltd. (Tokyo, Japan). Epididymal adipose tissue adiponectin levels were measured using ELISA kits (Proteintech Group, Inc., Tokyo, Japan).

### 4.4. Quantitative Reverse-Transcription Polymerase Chain Reaction (qRT-PCR)

RNA was extracted using the RNeasy Mini Kit (QIAGEN, Hilden, Germany), and cDNA was synthesized by reverse transcription using ReverTra Ace qPCR RT Master Mix (TOYOBO, Osaka, Japan), according to the manufacturer’s instructions. RT-PCR assay was conducted using 5 μL of cDNA as the template and 10 μL of Power SYBR Green PCR Master Mix (Thermo Fisher Scientific, Tokyo, Japan) on the Stratagene Mx 3005P QPCR System (Agilent Technologies, Santa Clara, CA, USA). The primers are listed in Table 1. Data were analyzed based on the 2^−∆∆Ct^ method and normalized by GADPH expression. The qPCR amplification was performed with an activation step at 95 °C for 10 min, followed by 40 cycles at 95 °C for 15 s (denaturation) and 60 °C for 1 min (annealing), and a dissociation stage at 95 °C for 15 s, 60 °C for 30 s, and 95 °C for 15 s for each gene.

### 4.5. Western Blot Analysis

Western blot analysis was performed using antibodies that specifically recognize proteins, including Glut4 and glyceraldehyde-3-phosphate dehydrogenase (Gapdh). The epididymal adipose tissue was homogenized, proteins were extracted, and 15 μg of extracted protein was loaded for sodium dodecyl sulfate–polyacrylamide gel electrophoresis immunoblot analysis. Protein bands were then transferred to polyvinylidene fluoride membranes (Bio-Rad Laboratories, Hercules, CA, USA). After blocking the nonspecific sites, the membrane was probed with primary antibodies, followed by a horseradish peroxidase-conjugated secondary antibody (Cell Signaling Technology, Inc., Danvers, MA, USA). Detection of antibody reactions was performed with ECL Western blotting Detection Reagents (Advansta, San Jose, CA, USA). Each band was normalized using the corresponding value of Gapdh as an internal control. The antibodies used were Gapdh (primary antibody (mouse monoclonal, Abcam, Cambridge, UK) 1:4000 dilution, secondary antibody (rabbit polyclonal, Abcam) 1:4000 dilution) and Glut4 (primary antibody (rabbit monoclonal, Cell signaling) 1:1000 dilution, secondary antibody (rabbit polyclonal, Abcam) 1:4000 dilution). Reaction times were overnight at 4 °C for primary antibodies and 1 h at room temperature for secondary antibodies.

### 4.6. Cell Culture

Mouse embryo 3T3-L1 cell line was obtained from the American Type Culture Collection (Manassas, VA, USA). 3T3-L1 pre-adipocytes were cultured in Dulbecco’s modified Eagle medium (Wako) supplemented with 10% fetal bovine serum (Sigma-Aldrich, St. Louis, MO, USA) at 37 °C in 5% CO_2_. AdipoInducer Reagent (Takara Bio, Otsu, Japan) was used for the differentiation of 3T3-L1 pre-adipocytes. For the differentiation of 3T3-L1 pre-adipocytes to mature adipocytes, 3T3-L1 pre-adipocytes were induced with differentiation media (DMEM with low glucose content supplemented with 10% fetal bovine serum, 2.5 μM dexamethasone (DEX), 0.5 mM 3-Isobutyl 1-methylxanthine (IBMX), and 10 μg/mL insulin (days 0–2). On day 2, the medium was replenished with maturation media (DMEM with high glucose content supplemented with 10% fetal bovine serum and 10 μg/mL insulin) and maintained at a 37 °C and 5% CO_2_ environment. This medium was changed every 2 days until day 8. At this time, the cells exhibited characteristics of mature adipocytes. At day 8, the medium was replenished, and 3T3-L1 mature adipocytes were treated with or without recombinant adiponectin (20 μg/mL, Prospec, Ness-Ziona, Israel). Samples were collected at 24 h to extract RNA after adiponectin treatment.

### 4.7. Statistical Analysis

All statistical analyses were performed using Ekuseru Toukei 2012 (SSRI, Tokyo, Japan). Data are reported as the mean ± standard error of the mean (SE). Statistical analysis was performed by Student’s *t* test. A difference was considered significant at *p* < 0.05.

## 5. Conclusions

This study showed for the first time that oral LPS administration has an inhibitory effect on type 2 diabetes mellitus. In addition, the study showed that LPS increases the expression levels of insulin signaling-related factors in adipose tissues, and the upregulator of these factors is adiponectin in adipose tissues. 

## Figures and Tables

**Figure 1 ijms-24-04619-f001:**
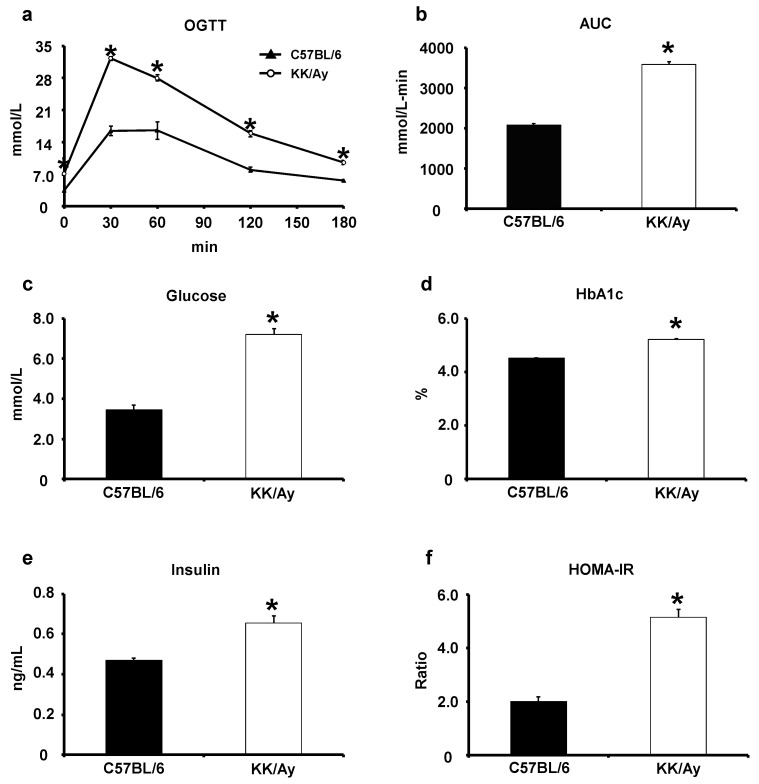
Comparison of biochemical parameters in C57BL/6 and KK/Ay mice. (**a**) An oral glucose tolerance test (OGTT) was performed (*n* = 5–6). (**b**) The AUC was calculated using the trapezoid rule (*n* = 5–6). (**c**) The fasting glucose was measured using commercial kits (*n* = 5–6). (**d**) The fasting HbA1c was measured by Oriental Yeast CO., Ltd. (*n* = 5–6). (**e**,**f**) Insulin levels and HOMA-IR were measured using commercial kits (*n* = 5–6). Values are presented as the mean ± SE. * *p* < 0.05 vs. C57BL/6 group.

**Figure 2 ijms-24-04619-f002:**
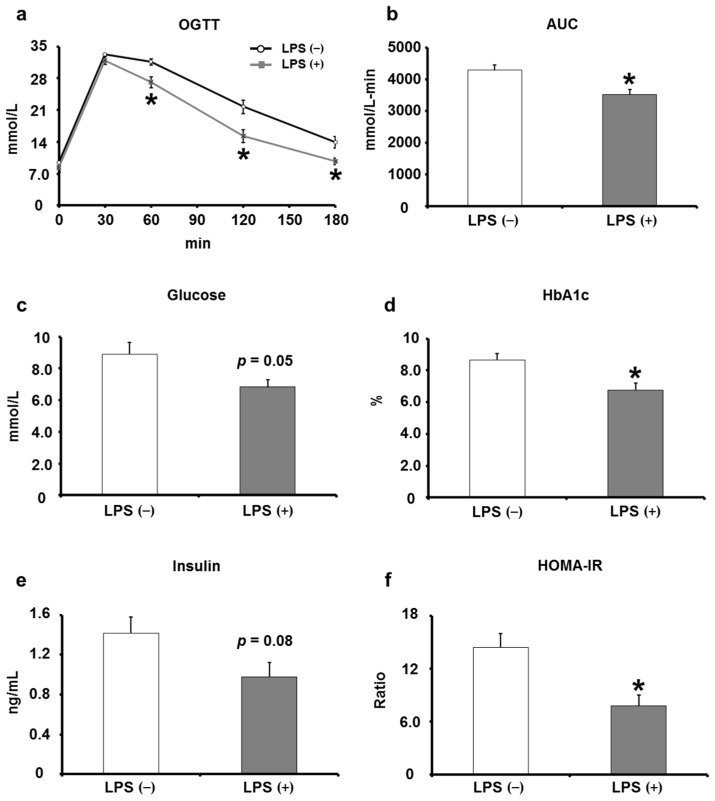
Effects of orally administered LPS on OGTT response, glucose, HbA1c, insulin, and HOMA-IR in KK/Ay mice. (**a**) OGTT was performed (*n* = 8). (**b**) The AUC was calculated using the trapezoid rule (*n* = 8). (**c**) The fasting glucose was measured using commercial kits (*n* = 5–6). (**d**) The fasting HbA1c was measured by Oriental Yeast CO., Ltd. (*n* = 5–6). (**e**,**f**) Insulin levels and HOMA-IR were measured using commercial kits (*n* = 5–6). Values are presented as the mean ± SE. * *p* < 0.05 vs. LPS (−) group.

**Figure 3 ijms-24-04619-f003:**
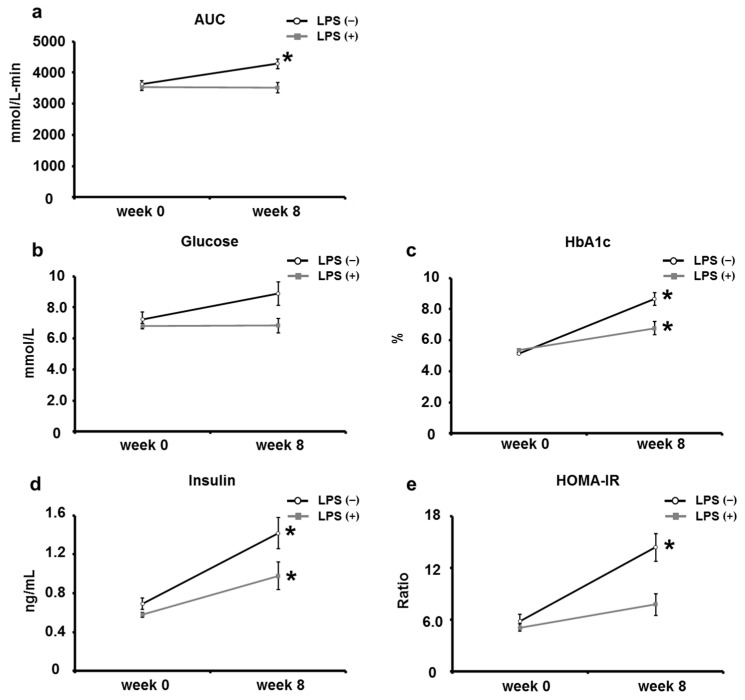
Comparison of OGTT response, glucose, HbA1c, insulin, and HOMA-IR before (week 0) and after (week 8) oral LPS administration. (**a**) The AUC of OGTT was calculated using the trapezoid rule (*n* = 5–8). (**b**) The fasting glucose was measured using commercial kits (*n* = 5–6). (**c**) The fasting HbA1c was measured by Oriental Yeast CO., Ltd. (*n* = 5–6). (**d**,**e**) Insulin levels and HOMA-IR were measured using commercial kits (*n* = 5–6). Values are presented as the mean ± SE. * *p* < 0.05 vs. mice before oral LPS administration (week 0).

**Figure 4 ijms-24-04619-f004:**
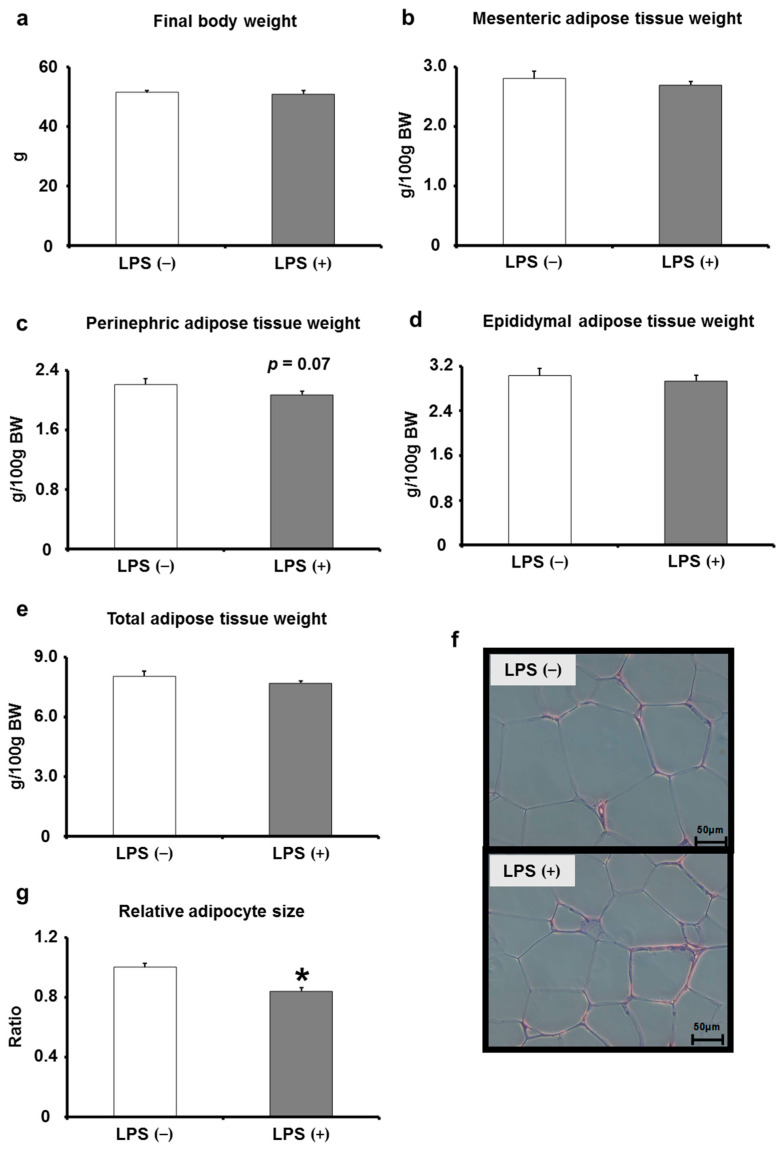
Effects of orally administered LPS on body weight, adipose tissue weight, and adipocyte size in KK/Ay mice. (**a**–**e**) The final body weight and mesenteric, perinephric, epididymal, and total adipose tissue weight were measured using an electronic analytical scale (*n* = 8). (**f**,**g**) The adipocyte size was measured using an optical microscope and analyzed by Image J *(n* = 8). Values are presented as the mean ± SE. * *p* < 0.05 vs. LPS (−) group.

**Figure 5 ijms-24-04619-f005:**
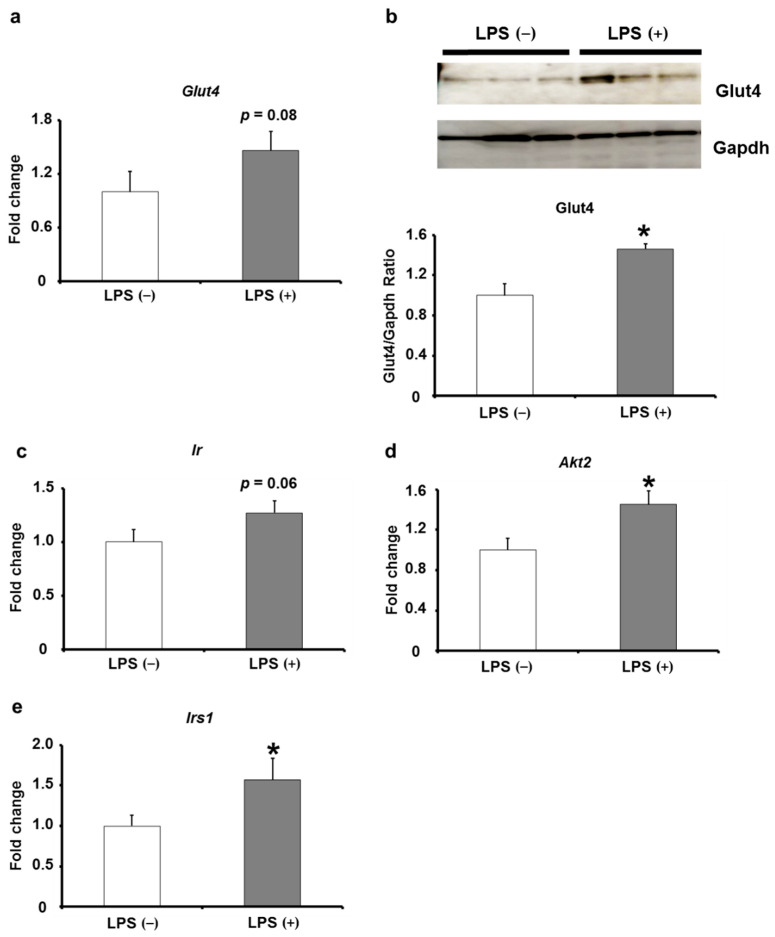
Effects of orally administered LPS on insulin signaling-related molecule expression in adipose tissues of KK/Ay mice. (**a**) The relative mRNA expression of *Glut4* was measured (*n* = 8). (**b**) The relative protein level of Glut4 was measured by Western blot analysis (*n* = 8). (**c**–**e**) The relative mRNA expression levels of *Ir*, *Akt2,* and *Irs1* were measured by quantitative reverse-transcription polymerase chain reaction (*n* = 8). Values are presented as the mean ± SE. * *p* < 0.05 vs. LPS (−) group.

**Figure 6 ijms-24-04619-f006:**
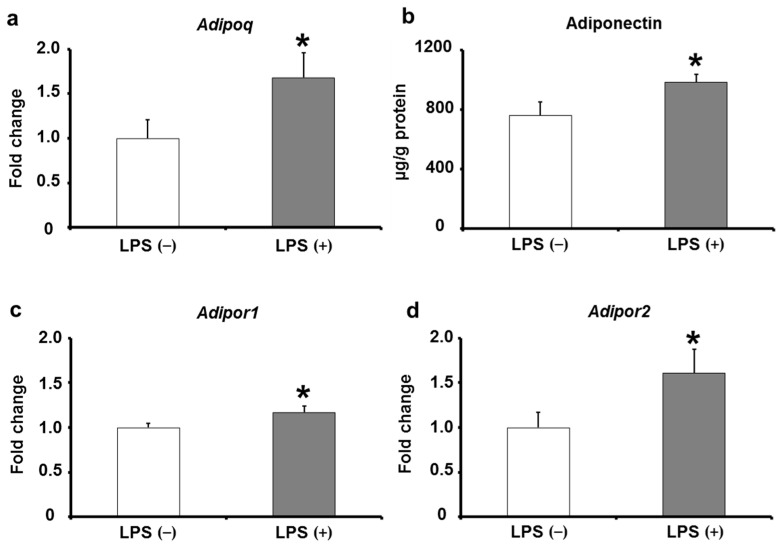
Effects of orally administered LPS on adiponectin expression in the adipose tissues of KK/Ay mice. (**a**) The relative mRNA expression of adiponectin was measured (*n* = 8). (**b**) The relative protein level of adiponectin was measured by enzyme-linked immunosorbent assay (*n* = 8). (**c**,**d**) Relative mRNA expression levels of *Adipor1* and *Adipor2* were measured by quantitative reverse-transcription polymerase chain reaction (*n* = 8). Values are presented as the mean ± SE. * *p* < 0.05 vs. LPS (−) group.

**Figure 7 ijms-24-04619-f007:**
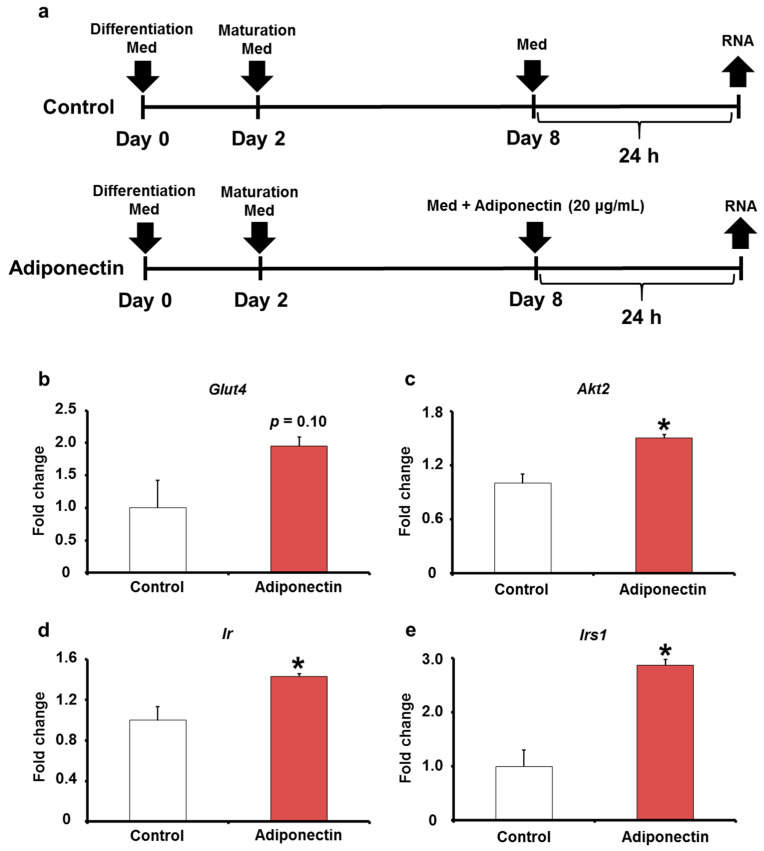
Effect of adiponectin on insulin signaling-related molecule expression in 3T3-L1 adipocyte. (**a**) Experimental design. (**b**–**e**) The relative mRNA expression levels of *Glut4*, *Ir*, *Irs1*, and *Akt2* in 3T3-L1 adipocyte were measured by quantitative reverse-transcription polymerase chain reaction (*n* = 3). Values are presented as the mean ± SE. * *p* < 0.05 vs. control group.

**Figure 8 ijms-24-04619-f008:**
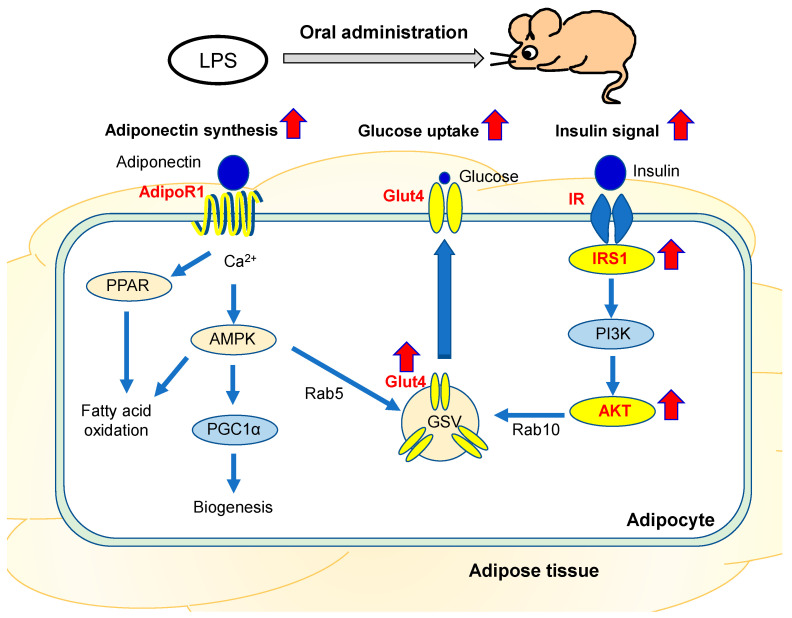
Scheme of signal transduction in adipose tissue during oral LPS administration.

**Table 1 ijms-24-04619-t001:** Primer pairs used for the quantitative RT-PCR analysis.

GenBank ID	Target Gene	Primer	Primer Sequence (5′–3′)
NM_009605	*Adipoq*	F	TGTTCCTCTTAATCCTGCCCA
		R	CCAACCTGCACAAGTTCCCTT
NM_001306069	*Adipor1*	F	AACGGGCCATCCATTTTTG
		R	TTAGCCGGGCTACATCAAGG
NM_197985	*Adipor2*	F	CCACACAACACAAGAATCCG
		R	CCCTTCTTCTTGGGAGAATGG
NM_001110208	*Akt2*	F	ACGTGGTGAATACATCAAGACC
		R	GCTACAGAGAAATTGTTCAGGGG
NM_001289726	*Gapdh*	F	CGACTTCAACAGCAACTCCCACTCTTCC
		R	TGGGTGGTCCAGGGTTTCTTACTCCTT
NM_001359114	*Glut4*	F	GTAACTTCATTGTCGGCATGG
		R	AGCTGAGATCTGGTCAAACG
NM_008361	*Il1b*	F	GAAAGACGGCACACCCACCCT
		R	GCTCTGCTTGTGAGGTGCTGATGTA
NM_031168	*Il6*	F	CCAGAGATACAAAGAAATGATGG
		R	ACTCCAGAAGACCAGAGGAAAT
NM_001303244	*Il12b*	F	ACAGCACCAGCTTCTTCATCAG
		R	TCTTCAAAGGCTTCATCTGCAA
NM_010568	*Ir*	F	TTTGTCATGGATGGAGGCTA
		R	CCTCATCTTGGGGTTGAACT
NM_010570	*Irs1*	F	CCATGAGCGATGAGTTTCGC
		R	GCAGTGATGCTCTCAGTTCG
NM_011333	*Mcp1*	F	AACTGCATCTGCCCTAAGGT
		R	ACTGTCACACTGGTCACTCC
NM_013693	*TNFα*	F	CTGTGAAGGGAATGGGTGTT
		R	GGTCACTGTCCCAGCATCTT

*Adipoq*, Adiponectin; *Adipor1*, Adiponectin receptor 1; *Adipor2*, Adiponectin receptor 2; *Akt2*, thymoma viral proto oncogene 2; *Gapdh*, glyceraldehyde 3 phosphate dehydrogenase; *Glut4*, glucose transporter type 4; *Il1b*, interleukin 1 beta; *Il6*, interleukin 6; *Il12b*, interleukin 12 subunit beta; *Ir*, insulin receptor; *Irs1*, insulin receptor substrate 1; *Mcp1*, Monocyte chemotactic protein 1; *TNFα*, tumor necrosis factor alpha.

## Data Availability

The authors confirm that the data supporting the findings of this study are available within the article and its Appendix A.

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
