# Peer review of "Oral Administration of Lipopolysaccharide Enhances Insulin Signaling-Related Factors in the KK/Ay Mouse Model of Type 2 Diabetes Mellitus"

_ijms, 2023, doi:10.3390/ijms24054619_

Round 1
Reviewer 1 Report
Yamamoto et al., found that oral administration of LPS suppressed the symptoms of type 2 diabetes, such as elevated of GTT, using KKAy mice. They showed that oral administration of LPS increased the expression of adiponectin and Glut4 in adipose tissue, which may contribute to the suppression of diabetic symptoms.
In addition, the in vitro experiments with 3T3 cells showed that adiponectin supplementation directly increased Ir, IRS, and AKT2. They concluded that oral administration of LPS is effective in type 2 diabetes, but the underlying mechanism for the effect of LPS on diabetes is still unknown.
This is an interesting observation and seems to be convincing. There are several issues that need to be addressed in the manuscript.
Comment
Fig1.
As they suggested in the text, KK/Ay mice are well known type 2 diabetes mellitus mice and many papers have reported impaired glucose tolerance and insulin resistance compared to control mice such as C57BL mice.
Therefore, the results they show in Figure 1 are not new and have only been checked for reproducibility for the previous papers.
They should not show these results as Figure 1 and may be able to show them in a supplementary figure. They can also refer to the previous papers in the text.
Fig 3
They write ‘The AUC of the OGTT increased significantly in the LPS (−) group at the end of the study compared with that at the start of the study’.
An alternative interpretation could be that OGTT increases without LPS administration. The results show that GTT increases when mice are 8 years old and that LPS administration suppresses this age-related increase in OGTT. I recommend that this explanation be modified.
Also, it would be easier for the reader to understand the aging process if the 0-8 weeks shown in Figure 3 were shown as the actual age of the mouse (e.g. 8, 16 weeks).
Discussion
They concluded that LPS affects adipose tissue, leading to suppression of type 2 diabetes.
They suggested the possibility that LPS acts on monocytes in adipose tissues.
Is TLR4 expressed in adipocytes? Is it possible that LPS affects macrophages more than monocytes?
There are a number of typographical errors throughout the manuscript.
Author Response
Thank you for providing these insights. We have improved the manuscript now. We trust this makes our manuscript acceptable now for publication in IJMS.
Point-by-point Response to Reviewer’s Comments:
[Comment 1]
As they suggested in the text, KK/Ay mice are well known type 2 diabetes mellitus mice and many papers have reported impaired glucose tolerance and insulin resistance compared to control mice such as C57BL mice.
Therefore, the results they show in Figure 1 are not new and have only been checked for reproducibility for the previous papers.
They should not show these results as Figure 1 and may be able to show them in a supplementary figure. They can also refer to the previous papers in the text.
[Response 1]
Thank you so much for your comments. We agree with you that Fig.1 is not a novel result. However, we believe that showing Fig.1 is necessary to support the credibility of the other results in this paper.
[Comment 2]
Fig 3 They write ‘The AUC of the OGTT increased significantly in the LPS (−) group at the end of the study compared with that at the start of the study’. An alternative interpretation could be that OGTT increases without LPS administration. The results show that GTT increases when mice are 8 years old and that LPS administration suppresses this age-related increase in OGTT. I recommend that this explanation be modified. Also, it would be easier for the reader to understand the aging process if the 0-8 weeks shown in Figure 3 were shown as the actual age of the mouse (e.g. 8, 16 weeks).
[Response 2]
Thank you for your suggestion. We agree that the alternative interpretation that you suggested is more understandable to the reader. So we have added the following sentence after the quoted sentence in the paragraph: 'This figure shows that OGTT expression increases with age and LPS oral administration suppressed this escalation.'
Thank you for your advice regarding the notation of Fig.3. We use this notation because we believe that it is more important to emphasize the duration of LPS administration rather than the age of the mouse. Therefore, we would like to keep the figure labeling to show the time before and after oral LPS administration. But, we have updated the labeling of the figure from ‘0wk & 8wk’ to ‘week 0 & week 8’, and the first sentence of Fig. 3 notation to 'Comparison of OGTT response, glucose, HbA1c, insulin, and HOMA-IR before (week 0) and after (week 8) oral LPS administration.' Hopefully, this makes it easier for the reader to understand that the number of weeks in the figure label refers to the duration of LPS administration, not the age of the mice.
[Comment 3]
Discussion
They concluded that LPS affects adipose tissue, leading to suppression of type 2 diabetes.
They suggested the possibility that LPS acts on monocytes in adipose tissues. Is TLR4 expressed in adipocytes? Is it possible that LPS affects macrophages more than monocytes?
There are a number of typographical errors throughout the manuscript.
[Response 3]
We appreciate the reviewer’s comments. Yes, TLR4 is also present in the adipocytes. Although we have not confirmed the mechanism by which LPS acts on adipose tissue, we believe it does not act directly on the tissue to enhance the expression of insulin signaling-related molecules. Since other groups have reported the release of proinflammatory cytokines and other effects on adipose tissue, we think that it is more likely that LPS acts indirectly on macrophages in adipose tissue.
Thank you for pointing it out. We have rechecked for typographical errors.
Reviewer 2 Report
In this study, the authors investigated the effect of oral lipopolysaccharide (LPS) administration on type 2 diabetes mellitus using the KK-Ay mouse model, which is the standard animal model of type 2 diabetes mellitus. The authors concluded that oral LPS
administration suppressed type 2 diabetes mellitus by inducing adiponectin expression
in adipose tissues and increasing the expression levels of insulin signaling-related factors.
Comments
The reviewer has some concerns as follows:
1. This is an interesting study. However, the safety for oral LPS to human is a serious concern. The relevant studies for oral LPS listed by the author in this manuscript appears to be the works of the author's research team (references #8-12). The safety and toxicological concerns for oral LPS application to humans should be discussed in detail. Moreover, the dosage of oral LPS is an important issue. Diabetes treatment is a long-term matter, and whether the dose used in animal experiments is toxic when used for a long time? It still needs to be considered and clearly discussed.
2. In Figure 1-d and f and Figure 2-d and f, there are a large gap between the values for HbA1c and HOMA-IR in LPS(-) groups, these gaps may lead to wrong conclusions. It should be clearly checked and explained.
3. In Figure 5-b, the images for GLUT4 and GAPDH are not convincing. It is hard to determine the effect of LPS on GLUT4 protein expression. It should be revised.
4. The effects of orally administered LPS on insulin signal-related molecule expression in adipose tissues of KK/Ay mice are really weak and limited.
5. The in vitro results of adiponectin in 3T3-L1 adipocyte shown in Figure 7 cannot explain the in vivo effects of LPS on adipose tissue.
Author Response
Thank you for providing these insights. We have improved the manuscript now. We trust this makes our manuscript acceptable now for publication in IJMS.
Point-by-point Response to Reviewer’s Comments:
[Comment 1]
This is an interesting study. However, the safety for oral LPS to human is a serious concern. The relevant studies for oral LPS listed by the author in this manuscript appears to be the works of the author's research team (references #8-12). The safety and toxicological concerns for oral LPS application to humans should be discussed in detail. Moreover, the dosage of oral LPS is an important issue. Diabetes treatment is a long-term matter, and whether the dose used in animal experiments is toxic when used for a long time? It still needs to be considered and clearly discussed.
[Response 1]
Thank you so much for your comments. In a previous study, our group has discovered that LPS can be found in the food that we eat, for example brown rice (The link to the journal paper: https://ar.iiarjournals.org/content/anticanres/36/7/3599.full.pdf). Furthermore, LPS derived from Pantoea agglomerans, the strain used in this study, has received GRAS and NDI certifications for toxicity related to oral administration, and is safe for animals to take orally for long periods of time. (The link to the journal paper: https://analyticalsciencejournals.onlinelibrary.wiley.com/doi/10.1002/jat.3987)
[Comment 2]
In Figure 1-d and f and Figure 2-d and f, there are a large gap between the values for HbA1c and HOMA-IR in LPS (-) groups, these gaps may lead to wrong conclusions. It should be clearly checked and explained.
[Response 2]
We appreciated the reviewer’s comment. Differences in HbA1c and HOMA-IR values in Figures 1 and 2 may be due to individual differences in the animals. We would like to emphasize that despite the differences, HbA1c and HOMA-IR values were significantly reduced in the group receiving orally administered LPS.
[Comment 3]
In Figure 5-b, the images for GLUT4 and GAPDH are not convincing. It is hard to determine the effect of LPS on GLUT4 protein expression. It should be revised.
[Response 3]
Thank you so much for your comment. The signal result shown in this image is a typical reaction of anti-GLUT4 antibody. We considered the data on Glut4 to be very important and have analyzed it multiple times using Cell Signaling's Glut4 antibody for protein expression in addition to mRNA. We found a significant increase in Glut4 expression was observed in the LPS orally administered group and is presented here. The images we obtained also look similar to published data by other groups.
[Comment 4]
The effects of orally administered LPS on insulin signal-related molecule expression in adipose tissues of KK/Ay mice are really weak and limited.
[Response 4]
Thank you so much for your comment. Oral administration of LPS may have limited effects on the expression of insulin signaling-related molecules, but it may also have effects on other factors involved in diabetes, given the decrease in HbA1c and HOMA-IR.
[Comment 5]
The in vitro results of adiponectin in 3T3-L1 adipocyte shown in Figure 7 cannot explain the in vivo effects of LPS on adipose tissue.
[Response 5]
Thank you so much for your comment. Although the experiment with 3T3-L1 adipocytes is not a direct result, we believe it provides indirect proof, as oral LPS administration significantly elevated adiponectin and increased the expression of insulin signaling-related molecules.
Reviewer 3 Report
The article submitted for my review is innovative and cognitive. The authors have an exciting idea to administer LPS KK/Ay orally to mice.
The design and implementation do not raise my objections. The introduction is correct. There is a clearly defined goal. The study was planned correctly.
The presentation of the results is clear, and the figures - do not seem to bear the features of unauthorized manipulation; they are legible, and their number is correct.
Only Figure 7a - is incomprehensible to me (?), and I don't think it is explained by the sentence from the text of the article: "Therefore, to investigate whether adiponectin directly induces an increase in the expression levels of insulin signaling-related factors, we stimulated 3T3 -L1 adipocytes in vitro with adiponectin".
This is animal research - hence my question about the decision numbers of the Bioethics Committee; as a rule, it is not enough to write that there was consent in such a general way as in the article.
Conclusion - there are many repetitions of the same content here. They should be read by the authors and corrected. There is also no explanation of what is in the second part of the title; "without inducing adipose tissue inflammation."
On the other hand, the title seems to be a bit too long - but I don't insist on changing it - I give it to the authors' attention.
I suggest the authors prepare a summary figure/scheme that would include paths/mechanisms (?) that would allow a better understanding of the obtained results.
Also, a diagram of the research process - better than Fig 7a - would be an excellent supplement.
Abstract: There is no clarified aim, although it is clearly defined in the article, so it should be added.
It is also not grammatically correct. The authors should reread it:
sentences like: "We speculate that oral LPS administration may alter glucose metabolism; however, details are unknown." The authors don't seem to mean that they are speculating and the details are unknown" - that sounds bizarre.
The literature was selected correctly.
Author Response
Thank you so much for your helpful remarks. We have improved the manuscript now. We trust this makes our manuscript acceptable now for publication in IJMS.
Point-by-point responses to reviewers' comments:
[Comment 1]
This is animal research - hence my question about the decision numbers of the Bioethics Committee; as a rule, it is not enough to write that there was consent in such a general way as in the article.
[Response 1]
I appreciate you pointing this out. We added the sentences in Materials and Methods as follows “The animal experiments were reviewed and approved by the Animal Care and Use Committee of the Control of Innate Immunity CIP. This experiment was conducted according to the Law for the Humane Treatment and Management of Animals Standards Relating to the Care and Management of Laboratory Animals and Relief of Pain (Ministry of the Environment, Japan), the Fundamental Guidelines for Proper Conduct of Animal Experiments and Related Activities in Academic Research Institutions (Ministry of Education, Culture, Sports, Science and Technology, Japan), and the Guidelines for Proper Conduct of Animal Experiments (Science Council of Japan).”
[Comment 2]
Conclusion - there are many repetitions of the same content here. They should be read by the authors and corrected. There is also no explanation of what is in the second part of the title; "without inducing adipose tissue inflammation."
[Response 2]
Thank you for your suggestion. We have moved the graph showing that there was no inflammation in adipose tissue during oral administration of LPS into the supplementary figure. So we have deleted that part of the title and also because the title is long, as you suggested.
We have also deleted the repeated contents from the conclusion section.
[Comment 3]
I suggest the authors prepare a summary figure/scheme that would include paths/mechanisms (?) that would allow a better understanding of the obtained results.
[Response 3]
Thank you for your suggestion. We have added the schematic figure (Figure 8) in the updated version of the manuscript.
[Comment 4]
Abstract: There is no clarified aim, although it is clearly defined in the article, so it should be added.
It is also not grammatically correct. The authors should reread it:
sentences like: "We speculate that oral LPS administration may alter glucose metabolism; however, details are unknown." The authors don't seem to mean that they are speculating and the details are unknown" - that sounds bizarre.
[Response 4]
Thank you for your suggestion. We have corrected the sentence you mentioned and added the aim of the study in the abstract. The updated part of the abstract is as below: (For the complete abstract, please refer to the updated version of the full manuscript.)
Lipopolysaccharide (LPS), an endotoxin, induces systemic inflammation by injection and is thought to be a causative agent of chronic inflammatory diseases, including type 2 diabetes mellitus (T2DM). However, our previous studies found that oral LPS administration does not exacerbate T2DM conditions in KK/Ay mice, which is the opposite of the response from LPS injection. Therefore, this study aims to confirm that oral LPS administration does not aggravate T2DM and to investigate the possible mechanisms.
Round 2
Reviewer 2 Report
This revised manuscript can be accepted. No further comments.